# Sounds of Nature in the City: No Evidence of Bird Song Improving Stress Recovery

**DOI:** 10.3390/ijerph16081390

**Published:** 2019-04-17

**Authors:** Marcus Hedblom, Bengt Gunnarsson, Martin Schaefer, Igor Knez, Pontus Thorsson, Johan N. Lundström

**Affiliations:** 1Department of Forest Resource Management, Swedish University of Agricultural Sciences, 901 83 Umeå, Sweden; 2Department of Ecology, Swedish University of Agricultural Sciences, 750 07 Uppsala, Sweden; 3Department of Biological and Environmental Sciences, University of Gothenburg, 405 30 Gothenburg, Sweden; bengt.gunnarsson@bioenv.gu.se; 4Department of Clinical Neuroscience, Karolinska Institutet, 171 77 Stockholm, Sweden; martin.schaefer@ki.se; 5Department of Social Work and Psychology, University of Gävle, 801 76 Gävle, Sweden; igor.knez@hig.se; 6Division of Applied Acoustics, Chalmers University of Technology, 412 96 Gothenburg, Sweden; pontus.thorsson@akustikverkstan.se; 7Monell Chemical Senses Center, Philadelphia, PA 19104-3308, USA; johan.lundstrom@ki.se; 8Department of Psychology, University of Pennsylvania, Philadelphia, PA 191 04, USA; 9Stockholm University Brain Imaging Centre, Stockholm University, 106 91 Stockholm, Sweden

**Keywords:** stress, experiment, virtual reality, soundscape, bird song, noise

## Abstract

Noise from city traffic is one of the most significant environmental stressors. Natural soundscapes, such as bird songs, have been suggested to potentially mitigate or mask noise. All previous studies on masking noise use self-evaluation data rather than physiological data. In this study, while respondents (*n* = 117) watched a 360° virtual reality (VR) photograph of a park, they were exposed to different soundscapes and mild electrical shocks. The soundscapes—“bird song”, “bird song and traffic noise”, and “traffic noise”—were played during a 10 min recovery period while their skin conductance levels were assessed as a measure of arousal/stress. No significant difference in stress recovery was found between the soundscapes although a tendency for less stress in “bird song” and more stress in “traffic noise” was noted. All three soundscapes, however, significantly reduced stress. This result could be attributed to the stress-reducing effect of the visual VR environment, to the noise levels being higher than 47 dBA (a level known to make masking ineffective), or to the respondents finding bird songs stressful. Reduction of stress in cities using masking with natural sounds requires further studies with not only larger samples but also sufficient methods to detect potential sex differences.

## 1. Introduction

The health and well-being of an urban population are strongly influenced by the characteristics of the urban systems where factors such as density of houses, the presence of green spaces, urban heat island effects, population densities, traffic, air pollution, and noise pollution all have an impact [1]. However, when it comes to environmental stressors associated with disease, noise pollution has been identified as one of the most significant factors [2,3,4,5,6,7]. In larger European cities, half the population is likely to be exposed to noise levels that produce physiological and psychological stresses that affect health and well-being (The European Directive COM 2002/49/EC), a problem that has been predicted to increase, leading to even higher societal costs [8,9]. Whereas research into this topic has traditionally focused on mechanical ways to reduce noise, few studies have assessed whether other environmental factors, such as the presence of nature or green areas, might reduce stress by their mere presence.

Urbanization naturally leads to a decreased availability to urban green areas [10], the very urban green areas that promote general human health and stress recovery [11]. Urban green areas can generate cognitive, affective, and psychophysiological benefits, reducing stress and attention fatigue [11,12,13]. However, the literature vaguely describes the concept of “green” areas and how “green” areas are linked to human well-being and mechanisms of stress reduction [14,15].

Most literature on perceived and actual attention restoration and stress recovery as the result of exposure to nature has focused on visual stimuli [11]. However, few studies have examined how sound in outdoor environments, such as urban green areas, influences stress reduction. Evaluating outdoor environments (compared to indoor environments) for their psycho-acoustic properties is a non-trivial task due to interactions between various sound sources and other acoustical and non-acoustical factors [16]. Studies conducted in outdoor environments commonly adopt a theoretical framework of soundscapes, which includes the total acoustic environment in which respondents perceive a sound [16,17]. Earlier studies that define well-being in general (self-reported well-being) and not stress reduction per se have emphasized the interdependence of visual and acoustic stimuli [18,19]. Individuals respond negatively to the lack of non-visual natural stimuli when observing nature through a video-only feed [20]. For example, participants reported missing “the smells and sounds” of nature and described the setting as “too quiet”. This observation suggests that natural sounds may contribute to the restorative experience of being in nature, perhaps because natural sounds signify a living or vital natural environment. Similarly, several self-evaluation studies have found that traffic noise is considered more stressful than natural sounds [19,21].

Urban noise can be reduced by using barriers [22], by enhancing certain sounds through architectural design strategies, and by masking noise using other sounds [16]. The least studied strategy for reducing urban noise is perhaps the masking strategy, which uses common sound sources from outdoor environments, such as bird songs and traffic noise [23], rather than masking speech with speech [24]. Investigating these relationships may help urban planners understand how natural sounds influence human well-being and health. According to Hao et al. masking is a “hearing phenomenon through which soundscape characteristics are altered by the presence of interfering sound event(s)” [23]. They further emphasize the importance of the specific context, such as real-life sound environments linked to birds chirping at different times of day, bird density, and visibility of sound sources. Most studies evaluating the impact of non-natural sounds are based on sounds that residents consider annoying [16]. However, natural sounds such as bird songs increase positive perceptions as well as reduce stress [25,26,27]. Few studies have investigated the physiological effects of positive soundscapes (cf. [28,29]), and previous experiments assessing the effects of masking noise with natural sounds rely on self-evaluations rather than physiological responses [16,30,31].

In this study, we investigate the physiological effects (stress reduction) of masking noise with a natural sound using an experimental design. We conducted the study in a laboratory to avoid bias linked to possible self-evaluation confounding factors in complex soundscape environments, personal elements such as expectation, preconception, and familiarity, as well as the need to control stimuli [32]. By using a laboratory setup, we controlled the soundscape environment as well as the visual environment, which relied on a multisensory virtual reality (VR) 360° setup.

This study aims to determine whether natural sounds from birds, presented at an ecologically valid sound level, reduce physical stress by masking noise from urban traffic. We hypothesize that when respondents are exposed to a stressor (mild electrical shocks), the type of soundscape they are exposed to will have a differential effect on their stress recovery. We predict that natural sounds will have the greatest effect on stress reduction, masked sounds from natural sources will have the second greatest effect on stress reduction, and noise from traffic will have the least effect on stress reduction.

## 2. Materials and Methods

### 2.1. Participants

This between n-group study included 117 participants (73 females and 44 males). Each participant was exposed to one of three types of soundscapes (“bird song”, “traffic noise”, and “bird song and traffic noise”) and one visual environment (360° VR photograph of a Swedish urban park). That is, the experiment included one soundscape environment for each participant, who were pseudo-randomly assigned a soundscape based on the order of participant entry: “traffic noise” (*n* = 38, 25 women, mean age 27.2 years, SD = 5.4), “bird song and traffic noise” (*n* = 39, 21 women, mean age 26.8 years, SD = 6.2), and “bird song” (*n* = 40, 27 women, mean age 27.5 years, SD = 4.9). The following inclusion criteria were used: self-declared health, age between 18 and 50 years, normal to corrected eye-sight and hearing, self-declared as not being currently pregnant, and no prescription medication use. Before enrolment, all participants provided signed informed consent. All research activities were performed in accordance with relevant guidelines/regulations as well as approved by the regional ethical review board in Uppsala (*Etikprövningsnämnden*, Dnr: 2016/175).

### 2.2. Virtual Reality Photograph

The experiment included one 360° virtual reality (VR) photograph of an urban park in Uppsala (Figure 1). The 360° photograph was presented using a VR mask (Oculus Rift). A Samsung Gear 360 SM-R210 camera was used to photograph a common park setting with an expansive lawn, some trees, distant buildings, paths, and a distant road. The camera was placed on a tripod at a height of 1.75 m. The photograph purposely lacked any elements of major commercial signs and was taken early in the morning to avoid pedestrians. In total, respondents spent 13 minutes in this environment (three minutes during the stress-inducing phase and ten minutes during the relaxation phase).

### 2.3. Soundscapes

The soundscapes (auditory stimuli) were presented through headphones integrated in the VR mask. The respondents were placed in a separate room with a closed door to minimize outside sound from entering the room. This closed door provided access to another room where a laboratory assistant was located. The door leading out of this room was also closed during the experiment, providing yet another sound barrier. To provide a soundscape with high positive perception, we used bird songs (highest rated natural sound in [26] and [22]) and the same number of species and abundance as the highest rated bird song combination in Hedblom et al. [25]. Although abundance and species number (*n* = 7) were equal (same number of strophes during the 30 s) to Hedblom et al. [33], the species composition in this study resembled the species composition heard in an urban park (the previously mentioned article used an urban woodlands soundscape). The species included in the present soundscape were willow warbler (*Phylloscopus trochilus*), chaffinch (*Fringilla coelebs*), blackbird (*Turdus merula*), blue tit (*Cyanistes caeruleus*), European robin (*Erithacus rubecula*), common swift (*Apus apus*), and common wood pigeon (*Columba palumbus*). The bird songs were downloaded from open sources on the internet and mixed using Audacity 2.1.2. To make the sound more realistic, the sound of a slight breeze was also mixed into the soundscapes. The bird songs were played in stereo, mimicking the way they are heard in nature, and the songs were heard in the left and right ears at different strengths (resembling a bird either far or near). The same bird song combination was used for the conditions “bird song” and “bird song and traffic noise”.

The condition “traffic noise” was recorded in the vicinity of a Swedish road in the city of Mariestad, which has a traffic intensity of 8000–10,000 vehicles a day (*Akustikverkstaden*). The recording was done from an open space at a distance of 20 m from the road. Human voices and bird songs were removed using the software program Audacity 2.1.2, leaving only the noise of vehicles. The noise from the traffic varied, including noise peaks when motorcycles or trucks passed by and less noise when the flow of traffic was less (but never absent).

When played through the headphones, the sounds had an intensity variation during the exposure at the level of the participant’s ears: “traffic noise”—*L*_p_ = 50–62 dBA; “bird song”—*L*_p_ = 45–80 dBA; and “bird song and traffic noise”—*L*_p_ = 50–77 dBA. The sounds and their corresponding A-weighted sound were chosen to resemble a realistic soundscape as if a respondent were standing in the park hearing noise at a distance and birds nearby and distant.

### 2.4. Experimental Procedure

These multisensory experiments, conducted in a laboratory at Karolinska Institutet in Stockholm, were specifically designed for using soundscape and visual stimuli. Upon arrival, the respondents provided signed informed consent and were informed about the experiment. In addition, they were told that they would be exposed to a virtual reality (VR) environment, mild electrical shocks, and skin conductance measurements. As the experiments were a between-group design, each participant was exposed to only one visual stimulus and one soundscape. We used mild electrical shocks instead of stressful social situations [34] or movies [13] to elicit a mild physical stress response. After an initial shock of 0.5 amperes was delivered, subsequent shocks were increased gradually by 0.3 amperes. The respondent rated the experienced shock of each increase on a 0–10 scale: 1 = “does not feel at all” and 10 = “hurts”. When the respondent perceived a 7 (“uncomfortable but not painful”), the intensity was considered to be proper and that intensity was then used throughout the experiment. That is, the intensity of the electrical shock was determined by the individual. Electrical shocks were generated via a Powerlab system (ADInstruments, Boulder, CO, USA) at different time intervals (see below). The shocks were delivered through electrodes on the middle phalanx of the index and middle fingers on the non-dominant hand. Skin conductance levels (SCL) were measured throughout the whole experiment using Ag/AgCl electrodes on the index and middle fingers of the dominant hand. Data were acquired at a 1000 Hz rate and then filtered offline using a 0.1 Hz high-pass filter.

While experiencing the virtual park, the participants received five shocks in total. The first shock started after 30 s and the last after 150 s. After 150 s, no more shocks were used although the participants were not aware that the stress induction period was over. After 180 s, the sound started and continued for 780 s (in total, ten minutes in one soundscape).

### 2.5. Statistical Analyses

Skin conductance levels were recorded continuously for the full duration that the participants spent in the virtual environment. The first 30 s of recording (before the initiation of the shock stimuli) were used to obtain baseline measurements for each participant. All subsequent values were adjusted by the mean value of the first 30 s of the experiment, creating a “baseline-adjusted skin conductance value”. For each participant, an average of the SCL values was taken every 30 s to remove spontaneous fluctuations and to reduce the resolution of subsequent statistical analyses. This technique resulted in 21 unique data points for each participant. Individuals with a mean value (within each experimental phase) more than three standard deviations from the mean had their value, within the phase in question, replaced with the group mean to limit data skew yet remain in the dataset.

To assess statistical effects, analyses were performed within the R environment (R Core Team, 2018), we used non-parametric tests where assumptions of normal distributions and equal variance were not met, otherwise, parametric tests were used. First, we assessed whether our paradigm could induce stress responses by performing a one-sample, one-tailed Student’s *t*-test against 0 (no change) on mean SCL values within the stress period with all three groups merged. Second, we assessed whether the three soundscapes produced different recovery effects during the relaxation period using ANOVA with the between-group factor “soundscape” (i.e., “bird song”, “traffic noise”, and “bird song and traffic noise”) and with the dependent variable SCL change (difference between the mean of the first 30 s and the mean of the last 30 s of the relaxation period). Third, we assessed whether there were differences between the soundscapes in their ability to reduce stress over time by dividing the SCL levels across the full ten minutes of the relaxation period into four-time segments. To this end, we created mean SCL values for the following periods: 180–330 s, 330–480 s, 480–630 s, and 630–780 s. These mean SCL values within each of the four periods were then entered into a mixed-ANOVA with the three soundscapes (“traffic noise”, “bird song”, and “bird song and traffic noise”), creating a 3 × 4 ANOVA model.

We further tested whether there were differences between sexes in stress reduction using a similar approach as above with ANOVA (the difference between the mean of the first 30 s and the mean of the last 30 s of the relaxation period) and mixed-ANOVA (sexes and four periods).

## 3. Results

### 3.1. Stress Induction

First, we assessed whether our paradigm successfully induced a stress response among participants. As predicted, compared to baseline, there was a significant increase in SCL in response to the weak electric shocks—*t* (9.48); df = 116, *p* < 2.2e-16. This increase indicated that our paradigm successfully induced a weak but reliable physical stress response.

### 3.2. Soundscapes and Stress

We then assessed whether there was a difference between the soundscapes in their ability to reduce stress following the stress induction period. We found no difference in stress reduction, *F* (2,114) = 0.187, *p* = 0.83, between the three soundscapes, as indicated by ANOVA (Figure 2).

A mixed-ANOVA comparing four periods with that of the three soundscapes revealed that there was a significant effect of period across all three soundscapes. In other words, SCL were reduced over the recovery period independent of soundscape (*p* < 0.001) (Table 1) (Figure 3). There were, however, no differences between the soundscapes (*p* = 0.181) nor an interaction between period and soundscapes (*p* = 0.995).

### 3.3. Sex and Stress

We found no significant differences (except for a decrease of stress) between the sexes, but, surprisingly, we found a tendency of differences between sexes during the relaxation period (Figure 4).

During the stress period (30–180 s), there was a significant difference in stress between sexes (Figure 4) as shown in the Student’s *t*-test based on the average SCL during the stress period, *t* (63) = 3.3021, *p* = 0.001575. The men chose slightly higher ampere values on average than women in the set up for the experiment (male average: 2.9; female average: 2.4). During the recovery, no differences were found in stress reduction between the sexes, *t* (87.42) = 1.6086, *p* = 0.1113.

There was stress reduction during the recovery period for both sexes (*p* < 0.001) (Table 2). When comparing stress recovery and sex for the four periods (1st: 180–330 s; 2nd: 330–480 s; 3rd: 480–630 s; and 4th: 630–780 s), there was a tendency of differences between sexes (*p* = 0.08523) (Table 2) although this was not statistically significant. No statistical significance was found between the interaction of sex and period.

## 4. Discussion

No statistically significant difference in stress reduction was found between the three soundscapes—“bird song”, “traffic noise”, and “bird song and traffic noise”. Although non-significant, there seems to be a pattern linked to our hypothesis that “traffic noise” would have the least reduction in stress and “bird song” would the greatest reduction in stress (Figure 2 and Figure 3). However, these results may be linked to the small sample size and potential large individual variation. Our hypotheses, however, were based on previous findings from studies that used self-evaluations to conclude that bird songs compared to other noises such as traffic noise reduce stress (cf. [16,35]). Understanding why our study found no statistically significant differences should help urban planners understand how to use natural sounds to mask urban noise to reduce noise-associated stress. Below, we discuss the importance of the context of the physiological responses to the soundscapes: (i) the link (interdependent) to the visual feature of urban green (in this case a park); (ii) masking sounds; (iii) dBA levels; (iv) demography, such as sex aspects and dose-response for the setting, for example, the time spent in the experimental setup; (v) the physiological tests of stress in the experimental set up; and (vi) the conclusions with recommendations based on results.

### 4.1. Interdependence of Soundscape and Visual Features

Overall, the induced stress decreased significantly during the ten minutes of the stress-recovery period irrespective of the soundscape. This decrease could be linked to visual features, such as urban green (or green in general) has been shown to be associated with stress reduction [36,37]. However, the link between sound and visual features are sometimes arbitrary and context dependent. In a self-evaluation experiment, Hong and Jeon [22] found that visual aspects were more important than auditory aspects of an experience. Other studies, however, such as Hedblom et al. [25], found that natural sounds of birds increase positive ratings of visual settings. Moreover, Viollon et al. [19] found that traffic noises and bird songs were significantly influenced by the visual degree of urbanization (self-evaluations). Thus, the results from Viollon et al. [19] support our results: a pleasant visual urban green environment increases the positive perception regardless of whether the soundscape contains traffic noise or bird songs, or both. We used a virtual reality photograph where the respondent was placed in the middle of an urban park with the possibility to view the scene from a 360° perspective, enabling the respondent to view the scene as if they were in the actual park, with the ability, for example, to look up trees, to look at the ground, and to look at the sky. This VR experience potentially provided a more visually relaxing (and realistic) environment than a conventional photograph of a green environment on a screen, which previous studies use to elicit responses, as VR increases the influence of visual features [19,25]. Nevertheless, it is not clear if it is the visual dominance of the VR photograph that provides the stress recovery or if people find all three sounds relaxing (possibly with bird songs being more relaxing than the other soundscapes) (Figure 3).

### 4.2. Masking of Sounds

In this study, we used bird songs to mask the noise because it is a common soundscape found globally in cities [38]. Typically, bird songs [23] and moving or running water [16,17,21,30,31] are the natural sounds used to mask noise. People’s perceptions of which sound is the most pleasant (or the most stress-reducing) vary between individuals and studies. In Hong and Jeon [22], bird songs are preferred to falling water, which was found to decrease the soundscape quality. Similarly, Ratcliffe et al. [26] found that bird songs were preferred to the sound of moving water; however, Jeon et al. [16] found that the sound of streams and lake waves were preferred over the sound of birds in a forest. Overall, soundscapes are rather vaguely defined and do not identify or describe the birds other than using the phrase “chirps of birds” although more than 10,000 bird species and even more bird songs exist. The type of species, abundance of species, and the combination of species affect people’s perceptions of an urban setting [25]. Furthermore, bird songs and water sounds are important features for masking noise when respondents are self-evaluating a soundscapes [21,35]. One explanation for the results in this study might be that differences detected in self-evaluations are not reflected in physiological responses; the differences are simply too small. Medvedev et al. [29] failed to reveal statistical differences between soundscapes and physiological responses, although not for self-evaluations. Therefore, it might be that subjective self-evaluation, or self-perception, of soundscapes are perceived as strong differences while actual physiological effects are not very large, at least not in the context of the soundscapes presented in this study (and in [29]). However, there might be a rather large individual variation in perception of a soundscape that impacts the overall results. Future studies should include recovery periods without any stimuli as well as only visual or only sound stimuli.

### 4.3. dBA Levels

Numerous studies assessing masking emphasize the importance of context where the dBA level is important [21,23]. Urban planning should consider specific dBA levels because individuals exposed to high noise levels commonly suffer from poorer health, including poor cardiovascular effects [39]. Regardless of dBA levels, respondents perceive an environment as more natural if they can hear bird songs [23]. Nonetheless, small differences have shown to influence self-evaluated perceptions of the masking effects that natural sounds have on noise. When traffic noise is lower (*L*_eq_ < 52.5 dBA, e.g., when the traffic noise is greater than 19 m away), the masking effects are more significant than if the noise is louder or the distance is shorter [23]. The results from Hao et al. [23] might contribute to our non-significant findings because our span of sound level for road traffic noise was as high as 62 dBA, a level that reduces the potential for bird song masking. Hao et al. [23] indeed show that when traffic noise exceeds 47.5 dBA, the pleasantness decreases sharply. As for bird songs in a real city park situation, the sound level of the bird songs is often low compared to the traffic. In this study, we wanted the bird songs to be audible in the mixed sound, so there is a risk that some of the bird song strophes reached an unrealistic level of 80 dBA on some occasions. A sound at this level can be a stressor due to its strength [40] rather than its subjectively interpreted content. This is something that should be tested in future experiments.

### 4.4. Demography and Time of Experiment

We found some tendencies of differences between sexes. Men showed higher stress initially during the stress period, a finding that is confirmed in other studies [41]. During the recovery period, there was a tendency that women, compared to men, had greater reductions in stress (*p* = 0.0853), but no significant difference was found. After eight minutes, women seemed to continue to experience stress reduction, whereas men seemed to increase stress (Figure 4). Previous experiments found that participants experience two minutes of stress reduction and four minutes of stress recovery. For example, it is important to study longer exposure times to include ecological validity and habituation effects [29]. We hypothesize that the tendency of men to experience a reduced recovery can be linked to previous results showing that men have less appreciation for urban green areas than women [42]. Previous studies of soundscapes reveal differences between the sexes where women, more than men, preferred natural sounds from birds [43]. Thus, a potential future replication of this study should provide a design that systematically assesses potential differences between men and women.

### 4.5. Measuring Stress in an Experimental Setup

We used electrical shocks to induce reliable physiological stress responses from all participants within an experimental setup and operationalized stress levels such as skin conductance levels (SCL). Other studies have used amylase enzyme [44] or heart rate [29] and each method has its advantages and disadvantages. How the physiological stress response is specifically linked to psychological or social stress is not known, although social stress seems to be one of the leading causes of impaired well-being in modern societies [45]. The direct link between environmental stimuli and long-term stress is difficult to assess using an experimental design without undue burden on the research participants. More studies are needed to fully understand the potential of natural sounds as masking sounds and their potential for reducing psychological stress.

## 5. Conclusions

We found no significant results for a reduction in physiological stress when masking traffic noise with bird songs. Stress levels during the recovery period were reduced below baseline levels for all soundscapes (although less so for traffic noise and least for bird songs), which could be due to the strong preferences for the visual 360° VR environment of a green park. It could also be that respondents, who were mostly urban dwellers, expected an urban park to have traffic noise and thus did not experience any impedance to their stress recovery from its presence. Our results further illustrate an example where bird songs do not have any physical effect on stress. Adding bird songs to a noisy traffic environment (above 47 dBA) might not reduce people’s stress but might lower noisy environments.

## Figures and Tables

**Figure 1 ijerph-16-01390-f001:**
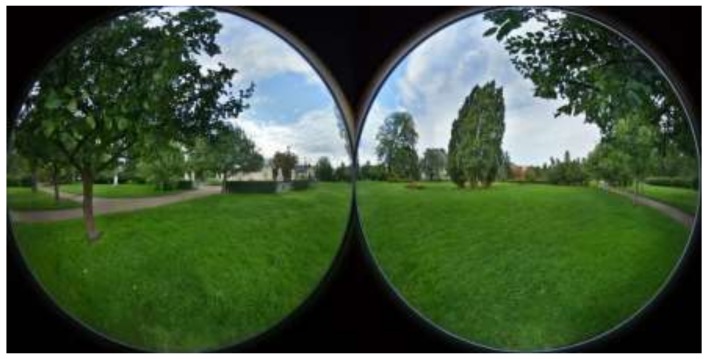
Illustrating the park photograph used in the experiment. The Oculus Rift provides a 360° view, but this photo only provides an approximation of the setting rather than how it would be perceived using the gear.

**Figure 2 ijerph-16-01390-f002:**
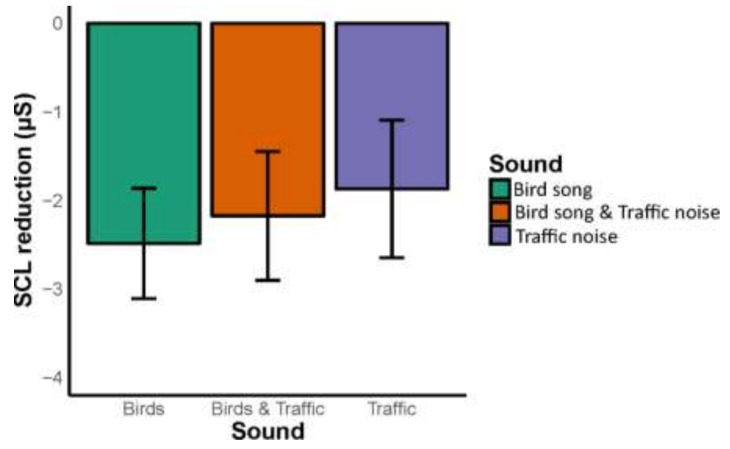
Average reduction change in skin conductance levels (μSiemens) during the relaxation period (from onset to end) for the three soundscape conditions. Errors bars indicate standard error of the mean (SEM).

**Figure 3 ijerph-16-01390-f003:**
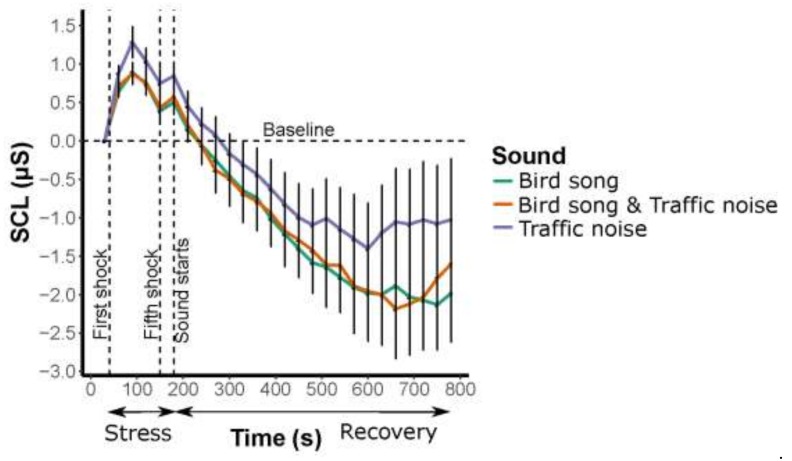
Skin conductance levels (SCL) responses (μSiemens) over time between the three soundscapes in the stress period (five minor electrical shocks) and the recovery period. Errors bars indicate the standard error of the mean (SEM).

**Figure 4 ijerph-16-01390-f004:**
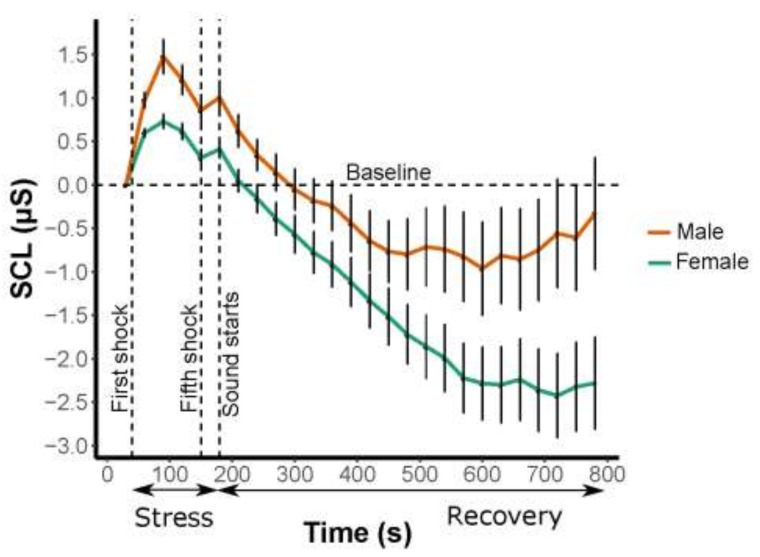
Illustrating non-significant differences in stress reduction (μSiemens) between male and female respondents during the stress period (five minor electrical shocks) and the recovery period (pooled values). Errors bars indicate standard error of the mean (SEM).

**Table 1 ijerph-16-01390-t001:** Mixed-ANOVA comparing soundscapes, recovery period (only soundscapes in the recovery period and no stress for ten minutes), and the combination of sounds and period.

	Df	Sum Sq	Mean Sq	*F* Value	Pr (>F)
Soundscape	2	33	16.25	1.715	0.181107
Recovery period	3	176	58.61	6.185	0.000399 *
Sound:period	6	6	1.04	0.110	0.995304
Residuals	456	4321	9.48		

* indicates a significant result.

**Table 2 ijerph-16-01390-t002:** Mixed-ANOVA comparing recovery period between sexes for ten minutes, four stress recovery periods, and the combination of sex and recovery period.

	Df	Sum Sq	Mean Sq	*F* value	Pr (>F)
Sex	1	26	25.63	2.973	0.085323
Recovery period	3	176	58.61	6.800	0.000171 *
Sex period	3	23	7.80	0.904	0.438825
Residuals	460	3965	8.62		

* indicates a significant result.

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
