# Peer review of "Sounds of Nature in the City: No Evidence of Bird Song Improving Stress Recovery"

_ijerph, 2019, doi:10.3390/ijerph16081390_

Reviewer 1 Report

important question - unique influence of sound (not sound and visual) on restoration. Comments below. None that should be too difficult to resolve.

Please provide line numbers for easier reviewing, also, I believe IJERPH uses brackets not footnotes for citations.

The claim that noise pollution is the most significatn factor probably should be tempered. I'd definitely want to see more than one reference here, and idally meta-analyses not a descriptive review (the systematic review here is on interventions not burden of disease). Also, this would likely vary considerably by locale.

Citation 3 is odd with the parenthese aftwerward, same with 4 being in the middle of a sentence.

The claim at the end of paragraph 1 doesn't quite work. Won't the policy implication be to change environmental factors to reduce noise?

Highly recommend supplementing current citation #9 with Taylor, L., & Hochuli, D. F. (2017). Defining greenspace: Multiple uses across multiple disciplines. Landscape and Urban Planning, 158, 25–28. http://doi.org/10.1016/j.landurbplan.2016.09.024

According to 15 is awkward as well

The Annerstadt paper may not be relevant here. Your premise is about noise pollution. This is about absence of noise in a forest spooking people out.

Lit review is missing important work on bird song restoration by Ratcliff:

Ratcliffe, E., Gatersleben, B., & Sowden, P. T. (2013). Bird sounds and their contributions to perceived attention restoration and stress recovery. Journal of Environmental Psychology, 36, 221–228. http://doi.org/10.1016/j.jenvp.2013.08.004

Ratcliffe, E., Gatersleben, B., & Sowden, P. T. (2016). Associations with bird sounds: How do they relate to perceived restorative potential? Journal of Environmental Psychology, 47(C), 136–144. http://doi.org/10.1016/j.jenvp.2016.05.009

Why is virtual capitalized in Virtual 360degree setup?

Why label treatment with traffic noise just "noise" and then the mixed as "traffic noise and bird song"? Please include SDs for ages here.

Inclusion criteria are appropriate, as is study design.

Samsung 360 camer is not a "Samsung 360 GEAR" and it is not 3d.

Oculus is mispelled as Occulus.

What was the sound environment of the  lab? Were the headphones noise canceling?

Duration of video, height of camera, etc should be reported.

Could you include experimental design figure? Difficult to translate into timeline with hundreds of seconds reported.

Results need a baseline test to compare across seamples. Also, it's unclear why you rea an t-test when you have 21? data points for SCL.

Regression (ANOVA) models need fixed effects of time.

What other individual differences / effect modification did you look at, beyond gender?

Figure 2 is nice and clear but confidence intervals (error bars) need to extend both up and down. Also, sound labels should match those used in the manuscript text.

Figure 3 is nice, should be higher resolution, again match labels and this would benefit from a timeframe figure as well.

Given the unique stressor, I think there should be a section on stressor types in lab exeperiments in the intro.

Also, it was not clear tha amperes was a DV as well, consider specifying meausures in a more organized (table?) format in the methods.

Did you qualitatively or quantitatively measure how much participants actually moved their head around in the enviornment? Can you provide a URL to the 360 degree video? Were all chairs used with casters/wheels?

SES in the conclusion seems to come out of the blue. As does sample size. You had >30 per group, did you run power analysis based on past work of effect size for electrical shocks?

Discussion in general is well presented and definitely well organized. Nice implications.

Author Response

REVIEWER 1

important question - unique influence of sound (not sound and visual) on restoration. Comments below. None that should be too difficult to resolve.

#1 Please provide line numbers for easier reviewing, also, I believe IJERPH uses brackets not footnotes for citations.

From authors: We have added line numbers for easier reviewing and changed the style to IJERPH with brackets.

#1 The claim that noise pollution is the most significatn factor probably should be tempered.

From authors: We added “…noise pollution has been identified as one of the most significant factors…”

#1 I'd definitely want to see more than one reference here, and idally meta-analyses not a descriptive review (the systematic review here is on interventions not burden of disease).

From authors: We totally agree and have removed the existing reference and added 6 additional references of which at least one is a meta-analyses.

Miedema, H. M. E.; Vos, H., Exposure-response relationships for transportation noise. J Acoust Soc Am 1998, 104, (6), 3432-3445.

3.         Munzel, T.; Gori, T.; Babisch, W.; Basner, M., Cardiovascular effects of environmental noise exposure. Eur Heart J 2014, 35, (13), 829-+.

4.         Raggam, R. B.; Cik, M.; Holdrich, R. R.; Fallast, K.; Gallasch, E.; Fend, M.; Lackner, A.; Marth, E., Personal noise ranking of road traffic: Subjective estimation versus physiological parameters under laboratory conditions. Int J Hyg Envir Heal 2007, 210, (2), 97-105.

5.         Babisch, W., Cardiovascular effects of noise. Noise Health 2011, 13, (52), 201-204.

6.         van Kempen, E. E. M. M.; Kruize, H.; Boshuizen, H. C.; Ameling, C. B.; Staatsen, B. A. M.; de Hollander, A. E. M., The association between noise exposure and blood pressure and ischemic heart disease: A meta-analysis. Environ Health Persp 2002, 110, (3), 307-317.

7.         An, R. P.; Wang, J. J.; Ashrafi, S. A.; Yang, Y.; Guan, C. H., Chronic Noise Exposure and Adiposity: A Systematic Review and Meta-analysis. Am J Prev Med 2018, 55, (3), 403-411.

 #1 Also, this would likely vary considerably by locale.

From authors: We agree, we discuss the importance of context and site on row 103-105; 342 and 377.

#1 Citation 3 is odd with the parenthese aftwerward, same with 4 being in the middle of a sentence.

From authors: We agree and have changed this.

#1 The claim at the end of paragraph 1 doesn't quite work. Won't the policy implication be to change environmental factors to reduce noise?

From authors: We agree and have changed this and removed text liked to policy, it was awkward.

#1 Highly recommend supplementing current citation #9 with Taylor, L., & Hochuli, D. F. (2017). Defining greenspace: Multiple uses across multiple disciplines. Landscape and Urban Planning, 158, 25–28. http://doi.org/10.1016/j.landurbplan.2016.09.024

From authors: Thank you for providing the reference, we added it into the text.

#1 According to 15 is awkward as well

From authors: We agree, we have put the reference in the end of the sentence instead.

#1 The Annerstadt paper may not be relevant here. Your premise is about noise pollution. This is about absence of noise in a forest spooking people out.

From authors: We agree, Annerstedt paper is removed. 

#1 Lit review is missing important work on bird song restoration by Ratcliff:

Ratcliffe, E., Gatersleben, B., & Sowden, P. T. (2013). Bird sounds and their contributions to perceived attention restoration and stress recovery. Journal of Environmental Psychology, 36, 221–228. http://doi.org/10.1016/j.jenvp.2013.08.004

Ratcliffe, E., Gatersleben, B., & Sowden, P. T. (2016). Associations with bird sounds: How do they relate to perceived restorative potential? Journal of Environmental Psychology, 47(C), 136–144. http://doi.org/10.1016/j.jenvp.2016.05.009

From authors: We agree, we added some text and also both references on Ratcliffe (of which the one from 2013 already was in the paper but in discussions). Now it reads: However, bird songs have showed to increase positive perceptions as well as reduce stress   [25-27].

#1 Why is virtual capitalized in Virtual 360degree setup?

From authors: We have now changed this to virtual 360° setup.

#1 Why label treatment with traffic noise just "noise" and then the mixed as "traffic noise and bird song"?

From authors: Good comment. We have changed this throughout the paper and in all figures as well.

#1 Please include SDs for ages here.

From authors: Information on SD is added.

#1 Inclusion criteria are appropriate, as is study design.

From authors: Thank you

#1 Samsung 360 camer is not a "Samsung 360 GEAR" and it is not 3d.

From authors: The camera used is named “Samsung Gear 360 SM-R210” so we changed that in the text. We removed 3D throughtout the text.

#1 Oculus is mispelled as Occulus.

From authors: We have changed this to Oculus. We further changed this under Figure 1 as well.

#1 What was the sound environment of the  lab? Were the headphones noise canceling?

Duration of video, height of camera, etc should be reported.

From authors: We added:

1) The camera was placed on a tripod at a height of 1.75 meters.

2) The respondents were placed in a separate room with closed doors where disturbances sound were minimal (outside the door was a second laboratory with one laboratory assistant which in turn also had a closed door). 

3) In total respondents spend 13 minutes in this environment (3 minutes during stress and 10 minutes during relaxation period).

#1 Could you include experimental design figure? Difficult to translate into timeline with hundreds of seconds reported.

From authors: We believe that adding yet another figure will not provide any substantial explanation to the seconds and have not added an experimental design figure.

#1 Results need a baseline test to compare across seamples. Also, it's unclear why you rea an t-test when you have 21? data points for SCL.

From authors: We respectfully disagree that we need to assess potential differences in baseline between groups. We standardize entry value by subtracting out the pre-shock values from all subsequent values (baseline correction) meaning that we are moving all participants into the same recording space. This remove the need to statistically assess potential differences since all starts from the value 0. Further, we performed t-tests because we averaged across time points. Either during the early/late period or over all.  

#1 Regression (ANOVA) models need fixed effects of time.

From authors: We argue that effect f “time” is accounted by the factor “period”, first as pre/post and later, as four time points.

#1 What other individual differences / effect modification did you look at, beyond gender?

From authors: We did not investigate other factors than what is mentioned in the manuscript.  

#1 Figure 2 is nice and clear but confidence intervals (error bars) need to extend both up and down. Also, sound labels should match those used in the manuscript text.

From authors: We agree and have changed this.  

#1 Figure 3 is nice, should be higher resolution, again match labels and this would benefit from a timeframe figure as well.

From authors: We agree and have changed this.

#1 Given the unique stressor, I think there should be a section on stressor types in lab exeperiments in the intro.

From authors: We partly agree, we decided to highlight the pros and cons about stressors in the discussion where it reads:

Measuring stress in an experimental setup

We used electrical shocks to induce reliable physiological stress responses for all participants within an experimental setup and operationalized stress levels as Skin Conductance Levels (SCL). Other studies have used Amylas [44]or heart rate [29]and each method has its pros and cons. How the physiological stress response is specifically linked to psychological or social stress is not known, although social stress seems be one of the leading causes of impaired wellbeing in modern societies [45]. The direct link between environmental stimuli and long-term stress is difficult to assess using an experimental design without undue burden on the research participants. To fully understand the potential in natural sounds, as masking effects and its potential for reducing psychological stress, further studies are needed.

#1 Also, it was not clear tha amperes was a DV as well, consider specifying meausures in a more organized (table?) format in the methods.

From authors: Ampere is not a dependent variable but merely a descriptive value (such as age) to inform the reader of what level of physical stimuli the participants were exposed to. This information was provided as reference for potential future replication but the factor of interest is the perceived irritation level, which was held constant between groups. Note that what ampere level produce a specific rated irritation level is highly variable between individuals.

#1 Did you qualitatively or quantitatively measure how much participants actually moved their head around in the enviornment? Can you provide a URL to the 360 degree video? Were all chairs used with casters/wheels?

From authors: No, we did not collect data on head movement or eye direction. The chair was not of the movable kind; it had 4 flat feet meaning that all exploration was done by a combination of head and eye motion. We could provide a link to an URL where the photo will be showed as a video. Supposedly this one: http://www.slu.se/en/departments/ecology/hemsidor/hedblom-marcus/ (however we have not done that yet but will provide the link prior to eventual publication of this paper and if the editors think this will be of importance).

#1 SES in the conclusion seems to come out of the blue. As does sample size. You had >30 per group, did you run power analysis based on past work of effect size for electrical shocks?

Discussion in general is well presented and definitely well organized. Nice implications

From authors:  We interpret SES as SE and have removed this from the conclusions. Unfortunately, to the best of our knowledge, this is the first study that use electric shock to stress individual within the context of relaxation towards nature perception. That excluded the possibility of a-priori power calculations. However, we recently performed a similar study (manuscript in revision) with a near identical setup but with more divergent/stronger manipulation of the nature experience where we found strong effects. We based our sample size on this study. Given that we in this study, with a much weaker manipulation, did not find an effect, it came natural to discuss a potential power problem.

Reviewer 2 Report

The study is interesting enough, but you refer to a great extend to the output of models without any dicussions of the implications of these results. You refer to numbers and differences in numbers without discussing the relevance of these differences. As an example paragraph 3.3: You refer to to an average current (for the shock) of 2.411321 A. Did you check the accuracy of your measuring equipment? Six digits after the decimal point give the impression of a very high accuracy, but it is doubtful if you could actually measujre microamperes.

You refer to different SCL during the recovery period. It would be a great improvement if you could also present the same data for a recovery period with no stimulation at all: no visual or audio. Or perhaps also just visual stimulation. This would show if the stimulation using masking sounds has any effect at all.

I would recommend a "washing" by an expert in the English language. The article is fully understandable, but the language can be improved.

Author Response

#2 The study is interesting enough, but you refer to a great extend to the output of models without any dicussions of the implications of these results.

From authors: We regret to say that we are not sure what the reviewer are referring to. The models we refer to in the text are our statistical models and the assumptions contained within them (one can view the term model as “statistical setup”). Moreover, in the discussion, we are not discussing our statistical models but rather the result and what implication these results have on our understanding of this complex phenomena.

#2 You refer to numbers and differences in numbers without discussing the relevance of these differences. As an example paragraph 3.3: You refer to to an average current (for the shock) of 2.411321 A. Did you check the accuracy of your measuring equipment? Six digits after the decimal point give the impression of a very high accuracy, but it is doubtful if you could actually measujre microamperes.

From authors: In respect of the level of electric stimulation, precise levels were assessed using the PowerLab measuring instrument where the sensitivity and accuracy is lower than what we report here. Nonetheless, we agree with the reviewer that this level of accuracy is not needed and we have limited the reported ampere to one decimal.

#2 You refer to different SCL during the recovery period. It would be a great improvement if you could also present the same data for a recovery period with no stimulation at all: no visual or audio. Or perhaps also just visual stimulation. This would show if the stimulation using masking sounds has any effect at all.

From authors: This is a good idea. Unfortunately, we do not have those data because it was not included in the design. We have now added a line in the discussion where we highlight that this would be a good future control. “Future studies should include recovery periods without any stimuli as well as only visual or only sound”.

#2 I would recommend a "washing" by an expert in the English language. The article is fully understandable, but the language can be improved.

From authors: We agree but due to the extremely short review time we did not have the possibility to send it to a professional editor. We let the editors decide if this is needed and if they then can prolong the time.  

Round  2

Reviewer 1 Report

Comments and concerns were well addressed

Author Response

Dear reviewer,

FROM REVIEWER: Thanks to the authors for revising their manuscript. Before endorsing its publication, I would like to ask them to address a few minor points.

FROM REVIEWER: In the cover letter, they mentioned about the figure and the professional copy-editing that reviewers suggested but there was no time to provide. I would like them to address these; please do ask for more time for revision if needed.

ANSWERS FROM AUTHORS: We got the extra time that was needed to make a professional copy-editing. Thanks for highlighting this to the editors.  

FROM REVIEWER: In particular, professional copy-editing might be beneficial as I have spotted a few typos myself (e.g., assesses instead of assessed; A-weight, instead of A-weighted; one acoustic stimuli instead of one acoustic stimulus - it is Latin indeed...), and also other aspects (e.g., I would use gender instead of sex, etc.).
ANSWERS FROM AUTHORS: An external professional editor did go through the whole text and hopefully it reads much better now.

As for “sex” versus “gender” we consider that “sex” is a term that refers to a biological construct whereas “gender” is a term that refers to a sociological construct. Our explanatory model is here a biological one, meaning that we believe that these effects originate from the participant’s biological sex and not their chosen gender identity. Thus, we argue that sex, rather than gender, is the correct label to use in this instance. Now, if the editor insist, we can change “sex” to “gender” but we then would insist on adding a sentence explaining that “gender” is used to adhere to the journal format and do not aligne with the correct use in the literature.

FROM REVIEWER: Please, revise with the support of an acoustics expert the units and weightings you reported about the magnitude/intensity of the auditory stimuli: did you refer to A-weighted equivalent sound levels (LAeq)? Please check carefully and revise if needed.

ANSWERS FROM AUTHORS: P. Thorsson in our team holds a PhD in Acoustics and changed the text accordingly. Yes, we refers to Laeq and thus the level range is the minimum and maximum levels evaluated with ’Fast’ time weighting.

FROM REVIEWER: You use the term soundscapes, but my feeling is that you should replace with "acoustic environments" instead. According to the ISO 12913-1 definition, soundscape is a "perceptual construct", while acoustic environments are actual physical phenomena. Thus, soundscapes "result" from acoustic environments (if people perceive them) but the terms are no equivalent. I would urge you to revise in all relevant occurrences in the text.

ANSWERS FROM AUTHORS: We agree that according to IS 12913-1 definition we should use “acoustic environment”. Nevertheless, we refer to the numerous literature that use Soundscape in a similar way as we did and argue that the use of soundscape of an environment of bird song is correct. See e.g.  Hume and Ahtabad 2013. “Physiological responses to and subjective estimates of soundscape elements” and Hao et al. 2016. “Assessment of the masking effects of birdsong on the road traffic noise environment”. We also highlight the link between soundscape and acoustic environment in the text: mention your suggestion in the sentence “Studies conducted in outdoor environments commonly adopt a theoretical framework of soundscapes, which includes the total acoustic environment in which respondents perceive a sound [16, 17]”.

FROM REVIEWER: Regarding the statistics, there is no mention about whether the assumptions for running a parametric test (i.e., ANOVA) were met: this might include normality, equality of variances, etc. If not, the strategy for statistical analysis should change and non-parametric alternatives should be sought.

ANSWERS FROM AUTHORS: We added “….we used non-parametric tests where assumptions of normal distributions and equal variance was not met, other vise, parametric tests were used….”. The whole sentence now reads: “To assess statistical effects, analyses were performed within the R environment (R Core Team, 2018); we used non-parametric tests where assumptions of normal distributions and equal variance was not met, other vise, parametric tests were used”.

Best regards